# Beyond Taylor Expansion: Intermediate Activation Perspectives in Structured Pruning

## Abstract

Extensive prior work on importance-based pruning relies on first- or second-order Taylor expansions of the loss to score parameters by the estimated loss increase upon removal. However, in large language models with massive parameters and multi-layered nonlinear mappings, such approximations inevitably lead to errors. When applied to structured pruning, Taylor-based criteria are typically extended from individual weights to entire neurons or channels by aggregating their sensitivities. While this enables parameter reduction at the structural level, Taylor expansion is constrained to low-order approximations, owing to the computational intractability of higher-order terms in large-scale models, which results in inaccurate estimates of loss change. Moreover, it neglects the hierarchical dependencies of deep models, failing to account for how parameters influence subsequent layers through forward propagation. In particular, the intermediate activations within the feed-forward network (FFN) layer provide a direct characterization of how the pre-activation projections transmits information forward, thereby offering a more faithful account of its influence on the model's representations. Therefore, we propose **ActTaylor**, an intermediate **act**ivation enhanced **Taylor** criterion for structured pruning, which integrates loss sensitivity with the hierarchical influence of parameters captured through intermediate activations. ActTaylor scores each hidden unit in the FFN by modulating its Taylor-based sensitivity with the activation statistics for one-shot pruning without any retraining. At pruning ratios of 20% and 30%, our method consistently outperforms state-of-the-art structured pruning baselines across seven commonsense benchmarks and one multi-task knowledge benchmark, improving the average accuracy on LLaMA-2 7B by 7.8% and 12.9%, and on LLaMA-2 13B by 12.5% and 14.0%, respectively.

## 1 Introduction

Large language models based on the Transformer architecture have achieved strong performance across many tasks in natural language processing, including question answering, summarization, and reasoning ((Wei et al., 2022)). Scaling laws suggest that increasing model size leads to consistent improvements in accuracy when sufficient data and computational resources are provided (Kaplan et al., 2020; Hoffmann et al., 2022). As a consequence, recent models often contain hundreds of billions of parameters, which results in high memory usage and computational cost during inference. These demands limit the practical deployment of large models in applications where computational efficiency is important. To address this issue, researchers have explored various methods of model compression, such as structural pruning(Ma et al., 2023a; Ashkboos et al., 2024), quantization (Frantar et al., 2023), and low rank approximation(Huang et al., 2025b), in order to reduce model size and improve inference efficiency without a significant loss in accuracy. This paper focuses on structural pruning, which specifically targets the feedforward layers of Transformer models. By removing the entire rows or column of the weight matrices, structural pruning significantly enhances inference efficiency and is particularly advantageous in hardware deployment. One key reason to focus on pruning the feed-forward networks (FFN) in Transformer architectures is that FFN layers account for a significant portion of the model's parameters and computational cost. Furthermore, prior work has shown that a small subset of FFN neurons with large activation norms dominate the model's inference behavior (Huang et al., 2025b).

Recent advancements in structural pruning have increasingly employed loss-aware importance criteria derived from Taylor expansions. Some approaches approximate the effect of parameter or channel removal on the global training objective by retaining first- or second-order terms of the Taylor expansion(Ma et al., 2023a; van der Ouderaa et al., 2024). Others, such as SlimGPT (Ling et al., 2024), adopt a more tractable approximation by applying Taylor expansion to a layer-wise reconstruction loss that serves as a proxy for the global objective. This allows them to capture not only the gradient but also the curvature of the loss landscape, providing a principled approximation of the parameter's contribution to overall model performance. While Taylor expansion provides a principled importance score for individual parameters, its extension to structured pruning requires additional considerations. At the neuron level, removing a unit in the FFN layer corresponds to eliminating an entire row of the input projection matrix together with the associated column in the output projection. In this setting, Taylor expansion evaluates the sensitivity of the loss to perturbations of the whole row vector of parameters. Although this row-level criterion is theoretically consistent, this criterion only captures the loss response and fails to reflect the neuron's actual forward contribution, which is directly manifested in its activation. Consequently, neurons with negligible activations that contribute little to the actual flow of information may still receive high Taylor scores. This discrepancy motivates us to further examine the role of activation statistics as indicators of how strongly a neuron participates in forward signal propagation. These considerations suggest that Taylor-based importance alone is insufficient to faithfully capture neuron relevance, and highlight the need for complementary measures that incorporate activation information.

To overcome this limitation, we propose ActTaylor, an activation-enhanced Taylor criterion for structured pruning. ActTaylor extends the classical Taylor expansion approach by incorporating moment-based statistics of intermediate activations as a measure of forward utilization. Each hidden unit in the FFN is assigned an importance score defined as the geometric mean of its Taylor-based sensitivity and its activation statistics, computed over calibration samples. This formulation captures both the estimated loss impact of parameter removal and the neuron's role in propagating information. Structured pruning is then performed by ranking units with this criterion and removing the least important ones. Results show that our one-shot pruning strategy is highly effective without requiring retraining, and that activation-enhanced Taylor expansion consistently outperforms Taylor-only pruning. Our contributions are as follows:

- We propose ActTaylor, an activation-enhanced Taylor criterion that remedies the limitations of classical Taylor expansion in structured pruning. By combining Taylor-based loss sensitivity with moment-based activation statistics through a geometric integration, ActTaylor provides a principled and faithful measure of neuron importance.

- ActTaylor enables one-shot pruning without any weight updates, compensation, or fine-tuning, making it both efficient and scalable for large language models.

- Experiments on LLaMA-2 and Mistral models demonstrate that ActTaylor consistently outperforms state-of-the-art structured pruning baselines. Across seven commonsense reasoning benchmarks and one multi-task knowledge benchmark, our method achieves superior performance compared to all baselines. In particular, it improves the average accuracy on LLaMA-2 7B by 7.8% and 12.9% at 20% and 30% sparsity, and on LLaMA-2 13B by 12.5% and 14.0%, respectively.

## 2 RELATED WORK

**Large Language Model Compression.** Large Language Models (Touvron et al., 2023a;b; Scao et al., 2023; Wang & Komatsuzaki, 2021; Achiam et al., 2023) refer to Transformer language models with billions or more parameters that consistently demonstrate outstanding performance across a wide range of tasks. However, their massive size and computational demands limit practical deployment, making model compression a key research direction. Existing approaches can be broadly categorized into quantization(Kim et al., 2023; Hooper et al., 2024), pruning(Frantar & Alistarh, 2023; Gao et al., 2024b; Ma et al., 2023b), knowledge distillation(Agarwal et al., 2023; Liu, 2024), and low-rank decomposition(Wang et al., 2025; Huang et al., 2025b). We concentrate on the pruning of the language models, especially the structural pruning. Pruning includes structural pruning, unstructural pruning, and semi-structural pruning. Unstructural pruning removes individual weights based on criteria such as magnitude, achieving high sparsity but lacking hardware efficiency(Frantar

& Alistarh, 2023; Xia et al., 2023; Sun et al., 2024). Semi-structural pruning enforces regular sparsity patterns (e.g., 2:4), balancing flexibility and hardware acceleration(Li et al., 2023; Sun et al., 2024; Frantar & Alistarh, 2023). Structural pruning eliminates entire neurons, layers, or heads, leading to direct speedup but risking larger accuracy loss(Ma et al., 2023b; van der Ouderaa et al., 2024; Gao et al., 2024b).

**Taylor Expansion-based Large Language Model Pruning.** Recent research has explored Taylor expansion-based pruning techniques for both structured and unstructured pruning of large language models. These methods approximate the impact of removing parameters by expanding the loss function around the current weights, using either first-order (gradient-based) or second-order (Hessian-based) information. GBLM-PrunerDas et al. (2024) exploits the first-order Taylor expansion, using normalized gradients from a small set of calibration samples in a training-free fashion to define its pruning criterion, and consistently surpasses methods such as SparseGPT(Frantar & Alistarh, 2023) and Wanda(Sun et al., 2024) across multiple benchmarks. LoRAPrune(Zhang et al., 2023) utilizes first-order information for pruning while employing LoRA-guided weights and gradients to approximate importance efficiently and facilitate rapid post-pruning recovery. LLM-PrunerMa et al. (2023b) compresses large language models in a task-agnostic manner by structurally pruning non-critical coupled structures with gradient information and efficiently recovering performance via lightweight LoRA tuning. Building on classical work in Optimal Brain Damage(LeCun et al., 1989) and Optimal Brain Surgeon (Hassibi et al., 1993), recent research incorporates second-order Taylor information to enable effective pruning of large language models. LLM Surgeon(van der Ouderaa et al., 2024) extends second-order pruning to large-scale LLMs by employing a Kronecker-factored Hessian approximation to estimate loss impact, computing optimal weight updates for compensation, and iteratively pruning with curvature updates to achieve high-accuracy compression. SlimGPT(Ling et al., 2024) performs structured pruning in a layer-wise manner, leveraging curvature-based error minimization with compensation, and employs batched greedy strategies plus grouped Cholesky decomposition to make Hessian-based pruning scalable to large LLMs. In summary, these studies all build on the idea of Taylor expansion to approximate the loss change caused by pruning and leverage various techniques—such as normalized gradients, LoRA-guided importance estimation, and Kronecker-factored Hessian approximations—to achieve more effective compression of large language models. In this work, we focus on structured pruning and propose a method that, grounded in Taylor expansion, further exploits the characteristics of neuron pruning by incorporating soft activation sparsity to address the limitations of Taylor-based importance estimation for neurons.

## 3 METHODOLOGY

We begin by formulating the problem of neuron pruning in Transformer feedforward networks. We then leverage Taylor expansion to approximate the loss change incurred by removing neurons, and analyze its limitations in capturing forward signal contributions. Motivated by this observation, we introduce an activation-enhanced Taylor expansion that combines loss sensitivity with activation statistics, leading to a more faithful measure of neuron importance.

### 3.1 NEURON PRUNING

In the Transformer architecture, each layer is composed of a multi-head self-attention (MHSA) module followed by a feedforward network (FFN). The FFN typically consists of two linear transformations separated by a non-linear activation function. For a given token representation $\mathbf{x} \in \mathbb{R}^d$, the FFN computes:

$$\text{FFN}(\boldsymbol{x}) = \boldsymbol{W}_2\, \sigma(\boldsymbol{W}_1 \boldsymbol{x}), \tag{1}$$

where $\boldsymbol{W}_1 \in \mathbb{R}^{d_{\text{ff}} \times d}$ and $\boldsymbol{W}_2 \in \mathbb{R}^{d \times d_{\text{ff}}}$ are weight matrices, and $\sigma(\cdot)$ is a non-linear activation function. In this work, we define the activation output $\boldsymbol{h} = \sigma(\boldsymbol{W}_1 \boldsymbol{x}) \in \mathbb{R}^{d_{\text{ff}}}$ as the set of FFN neurons. Each dimension of $\boldsymbol{h}$ corresponds to a neuron that captures a distinct non-linear transformation of the input token.

**Neuron Pruning** refers to the process of deactivating a subset of these neurons based on their importance. Structurally, this is equivalent to setting the corresponding rows in $\boldsymbol{W}_1$ and the corresponding

columns in $\boldsymbol{W}_2$ to zero, such that the pruned neurons no longer contribute to the output of the FFN. This structured pruning scheme reduces both the model's parameter count and computational cost, and it preserves the architecture's compatibility with efficient inference frameworks.

## 3.2 Taylor Expansion: Minimizing the Loss Change

In network pruning, our objective is to remove a subset of parameters while minimizing the performance degradation of a well-trained model. Let $\boldsymbol{W}^*$ denote the pretrained parameters obtained by minimizing the training loss $\mathcal{L}(\boldsymbol{W})$. We seek a masked parameter set $\hat{\boldsymbol{W}}$ such that the induced loss change is minimized under pruning constraints:

$$\hat{\boldsymbol{W}} = \arg\min_{\boldsymbol{W}} \left| \mathcal{L}(\boldsymbol{W}^* \backslash \hat{\boldsymbol{W}}) - \mathcal{L}(\boldsymbol{W}^*) \right|. \tag{2}$$

We focus on pruning neurons in the feed-forward network (FFN) layers of Transformer models. Consider the $\ell$-th FFN layer with input projection matrix $\boldsymbol{W}_\ell \in \mathbb{R}^{d_{\text{hid}} \times d_{\text{in}}}$. Deactivating a neuron $i$ corresponds to zeroing its associated row vector $\boldsymbol{w}_i$ in $\boldsymbol{W}_\ell$ and removing the matched column in the output projection.

Let $S$ denote a set of neurons in layer $\ell$, and $\hat{\boldsymbol{W}}_{\ell,S}$ the corresponding collection of row vectors. A second-order Taylor expansion of the loss around $\boldsymbol{W}^*$ restricted to layer $\ell$ gives:

$$|\Delta\mathcal{L}_{\ell,S}| \approx \left| \sum_{i \in S} \boldsymbol{w}_i^\top \nabla_{\boldsymbol{w}_i} \mathcal{L} - \frac{1}{2} \sum_{i,j \in S} \boldsymbol{w}_i^\top \boldsymbol{H}_{\ell\ell} \, \boldsymbol{w}_j \right|, \tag{3}$$

where $\boldsymbol{H}_{\ell\ell} \equiv \nabla^2_{\boldsymbol{W}_\ell} \mathcal{L}(\boldsymbol{W}^*)$ is the Hessian restricted to the parameters of layer $\ell$.

Directly computing the quadratic term is computationally intractable for large models. Instead, we approximate it using Hessian–vector products (HVPs), which allow efficient evaluation of the second-order contribution without explicitly forming the Hessian. This yields a per-neuron Taylor importance score:

$$\Delta\mathcal{L}_i^{\text{Taylor}} = \left| \boldsymbol{w}_i^\top \nabla_{\boldsymbol{w}_i} \mathcal{L} \right| + \frac{1}{2} \left| \boldsymbol{w}_i^\top \left( \boldsymbol{H}_{\ell\ell} \, \boldsymbol{w}_{\text{struc}} \right) \right|, \tag{4}$$

eq:per-neuron-bound where both terms are averaged over the calibration dataset and $\boldsymbol{w}_{\text{struc}}$ is a layer-wise representative direction. The detailed derivation of this formulation is provided in Appendix A.4.

## 3.3 Limitation of Taylor Expansion

**Setup.** Consider the FFN in a Transformer block with input $\boldsymbol{x} \in \mathbb{R}^d$. For the $i$-th neuron, let $\boldsymbol{w}_i^\top$ denote the $i$-th row of $\boldsymbol{W}_1$. Then the pre-activation, activation, and contribution to the output are

$$a_i = \boldsymbol{w}_i^\top \boldsymbol{x}, \qquad h_i = \sigma(a_i), \qquad z_i = (\boldsymbol{W}_2)_{:,i} \, h_i, \tag{5}$$

where $(\boldsymbol{W}_2)_{:,i}$ is the $i$-th column of $\boldsymbol{W}_2$. Thus each neuron $i$ is fully determined by its row vector $\boldsymbol{w}_i$ in $\boldsymbol{W}_1$ and propagates forward through the corresponding column of $\boldsymbol{W}_2$.

**Exact first and second derivatives.** For neuron $i$, the exact derivatives of the loss w.r.t. its row vector $\boldsymbol{w}_i$ are

$$\frac{\partial \mathcal{L}}{\partial \boldsymbol{w}_i} = \mathbb{E}\left[ \left( (\nabla_{\boldsymbol{h}} \ell)_i \, \sigma'(a_i) \right) \boldsymbol{x} \right], \qquad \frac{\partial^2 \mathcal{L}}{\partial \boldsymbol{w}_i \partial \boldsymbol{w}_i^\top} = \mathbb{E}\left[ \left( (\nabla_{\boldsymbol{h}} \ell)_i \, \sigma''(a_i) \right) \boldsymbol{x} \boldsymbol{x}^\top \right]. \tag{6}$$

Here, $\nabla_{\boldsymbol{h}} \ell$ denotes the gradient of the per-sample loss with respect to the hidden activations $\boldsymbol{h} = \sigma(\boldsymbol{a})$; $(\nabla_{\boldsymbol{h}} \ell)_i$ is the backpropagated signal associated with neuron $i$; and $\sigma'(\cdot), \sigma''(\cdot)$ are the first and second derivatives of the activation function.

**Taylor expansion vs. forward influence.** Consider one row $\boldsymbol{w}_i^\top$ of $\boldsymbol{W}_1$, which defines the pre-activation $a_i = \boldsymbol{w}_i^\top \boldsymbol{x}$ and the hidden activation $h_i = \sigma(a_i)$. A Taylor expansion of the loss $\mathcal{L}$ with respect to $\boldsymbol{w}_i$ around the current parameter value yields coefficients that are expectations of multi-linear forms in $\boldsymbol{W}_2^\top \nabla_{\boldsymbol{z}} \ell$, $\sigma'(a_i)$, $\sigma''(a_i)$, and so on, up to higher-order derivatives of $\sigma$. Importantly, these coefficients contain no monomial in the activation value $h_i$ itself.

In contrast, the forward contribution of neuron $i$ to the FFN output $\boldsymbol{z}$ is mediated directly by its activation:

$$\boldsymbol{z} = \sum_j (\boldsymbol{W}_2)_{:,j}\, h_j, \tag{7}$$

so the influence of neuron $i$ is proportional to $|h_i|$ through the term $(\boldsymbol{W}_2)_{:,i} h_i$. Thus the activation magnitude gates how much signal $\boldsymbol{w}_i$ actually transmits forward, whereas Taylor coefficients reflect only the sensitivity of the loss to perturbations of $\boldsymbol{w}_i$ and ignore this forward utilization.

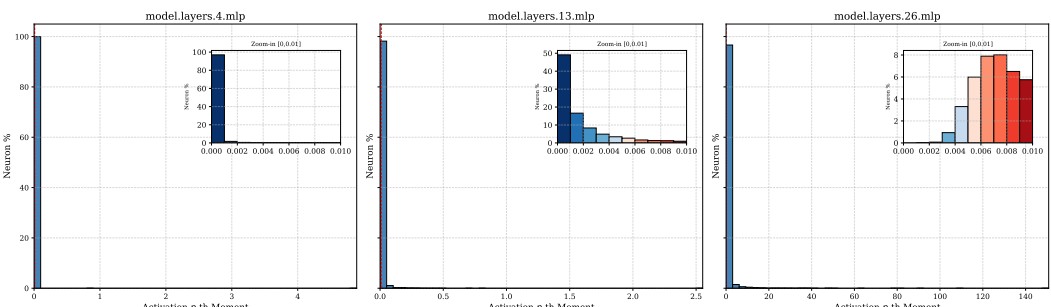

Figure 1: Distribution of the 4-th moment of neuron activations across different FFN layers of LLaMA-2. Shallow layers (e.g., Layer 4) show near-zero activations for almost all neurons. In middle layers (e.g., Layer 13), over half of the neurons remain close to zero, while deeper layers (e.g., Layer 26) display a broader distribution but still contain a large fraction of near-zero activations relative to their maximum values. This indicates that many neurons contribute little to forward propagation across depths.

**Illustrative example.** To see the discrepancy more concretely, consider the case of a ReLU activation $\sigma(a) = \max(0, a)$. Suppose $\boldsymbol{w}_i^\top \boldsymbol{x} < 0$. Then the corresponding activation is exactly zero: $h_i = \sigma(a_i) = 0$. In this case, setting $\boldsymbol{w}_i = 0$ leaves the hidden activation unchanged and therefore has no effect on the layer output $\boldsymbol{z}$ or on subsequent computations. However, the Taylor coefficients of $\mathcal{L}$ with respect to $\boldsymbol{w}_i$ need not vanish, because they depend on $\sigma'(a_i)$ and $\sigma''(a_i)$ evaluated at $a_i = 0$ and on the backpropagated gradient $(\nabla_{\boldsymbol{h}} \ell)_i$. Thus, Taylor expansion can assign non-trivial importance scores to parameters that have zero forward contribution, thereby overestimating their role.

This mismatch becomes especially problematic in large-scale models where sparse activations are prevalent. In Transformer FFNs, many neurons are inactive (i.e. $h_i \approx 0$) across most inputs(Figure 1). While the Taylor expansion emphasizes sensitivity of the loss to infinitesimal perturbations in $\boldsymbol{w}_i$, it ignores whether the neuron is actually utilized in the forward pass. Hence, neurons with persistently small activations may receive high Taylor scores despite contributing negligibly to the signal flow. Although state-of-the-art LLMs employ smooth gating activations such as SwiGLU or GeLU instead of ReLU, the same issue persists. These activations also yield many near-zero outputs due to their saturating nonlinearities. For such neurons, the contribution $(\boldsymbol{W}_2)_{:,i} h_i$ is trivial even if the associated Taylor coefficient is large. Moreover, while $\boldsymbol{W}_2$ could in principle amplify $h_i$, the near-zero activation ensures that the forward signal remains negligible in expectation over the dataset. Therefore, relying solely on Taylor expansion risks overvaluing neurons whose effective forward influence is absent.

In summary, Taylor expansion captures the local loss sensitivity to parameter perturbations but disregards the actual utilization of neurons in the forward pass. This limitation is exacerbated in over-parameterized models with highly sparse activations, where many neurons are rarely active and thus

play little role in practice. An accurate assessment of neuron importance should therefore integrate both perspectives: (1) the backward sensitivity encoded in Taylor coefficients, and (2) the forward contribution mediated by activation magnitudes. The next subsection develops such an activation-enhanced Taylor criterion.

### 3.4 ACTIVATION-ENHANCED TAYLOR EXPANSION FOR NEURON IMPORTANCE

The previous section has shown that Taylor expansion, while principled in capturing loss sensitivity, fails to account for the forward influence mediated by activation values. Building on this observation, we propose to augment Taylor-based neuron importance with activation statistics. By combining these two perspectives, our criterion provides a more faithful assessment of neuron importance for structured pruning. Concretely, we define a hybrid importance score that integrates both activation statistics and Taylor-based sensitivity through a geometric mean. Let $T_i$ denote the Taylor-based estimate of the expected loss increase incurred by removing neuron $i$, and let $A_i$ denote its $p$-th moment of activation over the calibration set. We then assign neuron $i$ the importance score

$$I_i = A_i^\lambda \cdot T_i^{1-\lambda}, \tag{8}$$

$$\text{where } \lambda \in [0,1], \quad A_i = \mathbb{E}_{\boldsymbol{x}\sim\mathcal{D}}[|h_i(\boldsymbol{x})|^p], \quad T_i = \mathbb{E}_{\boldsymbol{x}\sim\mathcal{D}}\left[|\Delta\mathcal{L}_i^{\text{Taylor}}(\boldsymbol{x})|\right].$$

Here $h_i$ denotes the activation of neuron $i$, $\mathcal{D}$ denotes the calibration dataset. $\Delta\mathcal{L}_i^{\text{Taylor}}$ denotes the loss perturbation from zeroing the parameter vector of neuron $i$, as defined in Eq. 4. The balancing coefficient $\lambda$ controls the relative emphasis on loss sensitivity versus activation statistics. The moment order $p$ determines which aspect of the activation distribution is captured, with smaller values emphasizing average magnitude and higher values emphasizing tail behavior. This multiplicative formulation ensures that a neuron is regarded as important only if it is simultaneously loss-sensitive and actively utilized in forward propagation. Unlike additive combinations, which allow one signal to dominate the other, the geometric mean enforces a balanced contribution, naturally penalizing neurons that score low on either dimension.

### 3.5 LAYER-WISE PRUNING RATIO

Determining the pruning ratio for each layer is a crucial step in structured pruning, as uniformly applying the same sparsity across all layers often leads to suboptimal results. Following prior work (Huang et al., 2025a), we adopt a linearly varying layer-wise sparsity schedule controlled by a single hyperparameter $\beta$. Formally, the sparsity ratio $s_i$ of the $i$-th layer is defined as:

$$s_i = S - \frac{\beta(L-1)}{2} + \beta \times (i-1), \qquad i = 1, 2, \ldots, L, \tag{9}$$

where $S$ denotes the target average sparsity across all layers, $L$ is the total number of layers, and $\beta$ controls the slope of sparsity allocation across depth.

## 4 EXPERIMENTS

### 4.1 EXPERIMENTAL SETTINGS

**Implementation details.** We use the first shard of the C4 dataset as the calibration set(Raffel et al., 2020), from which 256 random samples with a sequence length of 4096 tokens are selected to compute pruning criteria. The hyperparameters in the pruning criteria $\lambda$ are selected via grid search on calibration set C4. The moment order $p$ is set to 4 throughout our experiments. The Hessian Vector Product is computed on four NVIDIA H100 GPUs, while all other pruning and evaluation experiments are conducted on a single NVIDIA H100.[1]

**Models and Evaluation.** We evaluate our approach on several decoder-based generative models, including LLaMA-2 7B, 13B(Touvron et al., 2023b), and Mistral 7B-v0.1 (Jiang et al., 2023). Language modeling performance is measured by perplexity on the WikiText2 test set (Merity et al.,

---

[1]Our code is available at https://anonymous.4open.science/r/LLMPruning-52F0.

2016). The commonsense reasoning capability is evaluated on eight benchmarks: BoolQ, PIQA, HellaSwag, WinoGrande, ARC-easy, ARC-challenge, OpenBookQA, and MMLU. Among these, MMLU is evaluated in the 5-shot setting, while all others are assessed in the zero-shot setting. All evaluations are conducted using the LM Evaluation Harness framework(Gao et al., 2024a).

**Baselines.** To validate the superiority of our method over existing Taylor expansion–based approaches, we compare against LLM-Pruner(Ma et al., 2023b), the state-of-the-art structured pruning method built upon Taylor expansion. In addition, we include recent importance-based methods such as FLAP(An et al., 2024), as well as decomposition-based pruning approaches, including SVD-LLM(Wang et al., 2025) and SliceGPT(Ashkboos et al., 2024).

## 4.2 MAIN RESULTS

**Perplexity and Commonsense Reasoning Task Performance.** Table 1 reports the perplexity and downstream performance of our method and competing baselines on LLaMA-2 models (7B, 13B) and Mistral-7B under 20% and 30% structured pruning. The evaluation covers both generalization ability (measured by perplexity on WikiText2) and reasoning capability across seven commonsense benchmarks and one multi-task knowledge benchmark. Since the pruning strategy of LLM-Pruner is not applicable to Mistral 7B, its results are omitted in that part of the table. Across all models and sparsity levels, ActTaylor consistently achieves the best average performance. For LLaMA-2 7B, ActTaylor improves average accuracy by 7.8% at 20% pruning and 12.9% at 30% pruning over the best baseline. For LLaMA-2 13B, the gains are even larger, with 12.5% at 20% pruning and 14.0% at 30% pruning; notably, at 20% sparsity, the average accuracy of ActTaylor is only 3% lower than the unpruned model, demonstrating strong retention of performance under compression. On Mistral 7B, ActTaylor also surpasses all baselines, delivering strong improvements at both pruning ratios while maintaining competitive perplexity. Importantly, all these results are obtained in a one-shot pruning setting without any weight updates, compensation, or fine-tuning, highlighting the effectiveness and practicality of ActTaylor for compressing large-scale language models.

Table 1: Perplexity and downstream performance of LLaMA-2 models (7B, 13B) and Mistral-7B at 20% and 30% compression ratio. Best results are in **bold**.

| Model | Method | PPL↓ | Avg.↑ | MMLU | PIQA | BoolQ | WinoG. | HellaS. | ARC-e | ARC-c | OBQA |
|---|---|---|---|---|---|---|---|---|---|---|---|
| LLaMA-2 7B 20% | Original | 5.11 | 64.10 | 45.70 | 79.05 | 77.77 | 69.38 | 75.92 | 74.49 | 46.25 | 44.20 |
| | LLM-Pruner | 10.55 | 55.12 | 26.20 | 75.95 | 63.76 | 63.38 | 67.83 | 64.31 | 39.93 | 39.60 |
| | FLAP | 6.76 | 53.18 | 31.90 | 74.54 | 53.94 | 62.98 | 64.74 | 61.28 | 36.43 | 39.60 |
| | SliceGPT | 9.70 | 41.84 | 26.30 | 61.26 | 37.92 | 59.83 | 44.28 | 46.09 | 28.41 | 30.60 |
| | SVD-LLM | 7.84 | 46.73 | 26.80 | 65.13 | 54.68 | 62.43 | 51.73 | 47.22 | 27.82 | 38.00 |
| | **ActTaylor (Ours)** | **6.70** | **59.45** | **36.80** | **76.88** | **72.51** | **67.88** | **71.85** | **67.09** | **41.98** | **40.60** |
| LLaMA-2 7B 30% | LLM-Pruner | 18.25 | 47.67 | 24.60 | 72.25 | 53.24 | 54.54 | 56.96 | 51.09 | 31.66 | 37.00 |
| | FLAP | 8.91 | 48.93 | 26.70 | 70.29 | 52.20 | 60.06 | 56.58 | 55.18 | 32.25 | 38.20 |
| | SliceGPT | 15.42 | 37.57 | 25.90 | 55.55 | 37.83 | 54.46 | 35.17 | 39.06 | 24.57 | 28.00 |
| | SVD-LLM | 11.40 | 42.54 | 25.50 | 60.01 | 51.80 | 58.25 | 41.85 | 43.31 | 25.43 | 34.00 |
| | **ActTaylor (Ours)** | **8.49** | **55.25** | **31.70** | **74.10** | **67.83** | **65.27** | **66.54** | **60.27** | **37.71** | **38.60** |
| LLaMA-2 13B 20% | Original | 4.57 | 67.55 | 55.40 | 80.41 | 80.55 | 72.53 | 79.41 | 77.39 | 49.15 | 45.60 |
| | LLM-Pruner | 9.67 | 56.39 | 22.80 | 77.97 | 62.97 | 60.77 | 71.26 | 67.09 | 44.28 | 44.00 |
| | FLAP | 5.90 | 58.18 | 41.20 | 75.57 | 66.42 | 67.25 | 69.19 | 65.91 | 39.08 | 40.80 |
| | SliceGPT | 8.21 | 48.99 | 35.49 | 65.18 | 37.86 | 65.67 | 52.30 | 59.26 | 36.77 | 39.40 |
| | SVD-LLM | 6.18 | 57.90 | 35.54 | 72.91 | 72.17 | 68.43 | 63.47 | 71.00 | 39.93 | 41.00 |
| | **ActTaylor(Ours)** | **5.75** | **65.45** | **49.20** | **78.73** | **80.86** | **71.43** | **77.16** | **77.16** | **48.04** | **44.40** |
| LLaMA-2 13B 30% | Original | 4.57 | 67.55 | 55.40 | 80.41 | 80.55 | 72.53 | 79.41 | 77.39 | 49.15 | 45.60 |
| | LLM-Pruner | 12.47 | 50.93 | 22.80 | 73.18 | 62.11 | 57.93 | 60.89 | 54.71 | 34.04 | 41.40 |
| | FLAP | 7.08 | 54.29 | 33.20 | 72.42 | 64.37 | 63.93 | 62.44 | 61.45 | 37.29 | 39.20 |
| | SliceGPT | 12.68 | 39.54 | 27.10 | 56.75 | 37.83 | 57.70 | 38.27 | 40.87 | 26.19 | 31.60 |
| | SVD-LLM | 7.93 | 48.54 | 28.60 | 65.56 | 64.01 | 63.93 | 48.00 | 50.59 | 30.03 | 37.60 |
| | **ActTaylor(Ours)** | **7.05** | **61.91** | **45.50** | **76.66** | **76.39** | **68.75** | **73.38** | **68.31** | **44.28** | **42.00** |
| Mistral 7B 20% | Original | 4.92 | 70.12 | 62.50 | 82.05 | 83.98 | 73.95 | 81.02 | 79.55 | 53.92 | 44.00 |
| | FLAP | 7.11 | 50.03 | 25.90 | 72.31 | 62.26 | 64.09 | 55.94 | 51.05 | 31.91 | 36.80 |
| | SliceGPT | 8.23 | 42.73 | 28.60 | 60.66 | 37.86 | 59.43 | 45.10 | 48.15 | 30.03 | 32.00 |
| | SVD-LLM | 7.26 | 57.77 | 41.80 | 73.39 | 68.29 | 68.43 | 61.75 | 71.34 | 40.53 | 36.60 |
| | **ActTaylor(Ours)** | **6.63** | **64.70** | **52.90** | **80.03** | **75.93** | **69.46** | **75.36** | **72.77** | **47.35** | **43.80** |
| Mistral 7B 30% | FLAP | 13.10 | 49.61 | 26.40 | 69.59 | 65.26 | 64.80 | 55.61 | 48.91 | 30.55 | 35.80 |
| | SliceGPT | 14.69 | 35.77 | 25.00 | 54.41 | 37.83 | 51.62 | 32.54 | 35.02 | 22.95 | 26.80 |
| | SVD-LLM | 12.32 | 49.06 | 28.20 | 64.91 | 64.62 | 64.17 | 47.36 | 58.25 | 30.72 | 34.20 |
| | **ActTaylor(Ours)** | **8.46** | **58.54** | **43.60** | **76.66** | **66.85** | **66.85** | **69.44** | **64.35** | **41.21** | **39.40** |

These findings highlight that ActTaylor is a simple yet effective criterion for structured pruning, providing a strong balance between efficiency and performance across different model scales and pruning settings.

### 4.3 ABLATION STUDY

To better understand the effectiveness of our proposed ActTaylor criterion, we conduct ablation studies comparing it with the classical Taylor-based pruning method. As shown in Figure 2, across both LLaMA-2 7B and Mistral 7B, ActTaylor consistently achieves lower perplexity than Taylor expansion under the same pruning ratio. The performance gap becomes more pronounced as the pruning ratio increases, highlighting that activation statistics effectively mitigate the approximation error of Taylor expansion and provide more stable importance estimates. These results demonstrate that incorporating activation information is particularly beneficial in preserving language modeling capability when models are compressed to higher sparsity levels.

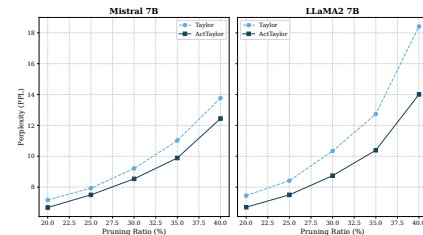

Figure 2: Perplexity comparison of Taylor vs. ActTaylor across pruning ratios on LLaMA-2 7B.

## 5 CONCLUSION

In this work, we introduced ActTaylor, an activation-enhanced Taylor criterion for structured pruning of large language models. By combining loss sensitivity with activation statistics, our method addresses the intrinsic limitations of Taylor expansion in capturing neuron importance. We developed a principled formulation that integrates both perspectives through a multiplicative scheme, supported by a theoretical justification and efficient approximations via Hessian–vector products. Extensive experiments across multiple model scales and benchmark tasks demonstrated that ActTaylor achieves more favorable trade-offs between compression and performance compared to existing approaches. Our study highlights the importance of incorporating forward activation information into pruning criteria and provides a general framework that can be extended to future model compression techniques.

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

# A   APPENDIX

## A.1   ETHICS STATEMENT

This work adheres to the ICLR Code of Ethics. No human subjects or animal experimentation were involved in this study. All datasets used, including the epigenomic histone modification dataset and the plant core promoter dataset, were sourced in compliance with relevant usage guidelines, ensuring no violation of privacy. We took care to avoid biases or discriminatory outcomes in our research process. No personally identifiable information was used, and no experiments were conducted that could raise privacy or security concerns. We are committed to maintaining transparency and integrity throughout the research process.

## A.2   REPRODUCIBILITY STATEMENT

We have made every effort to ensure that the results presented in this paper are reproducible. All code and datasets will be made publicly available to facilitate replication and verification upon publication. The experimental setup, including training steps, model configurations, and hardware details, is described in detail within the paper.

The datasets employed—C4 and Wikitext2—are publicly accessible, ensuring consistency and reproducibility of evaluation results.

We believe these measures will enable other researchers to reproduce our findings and further advance the field.

## A.3   LLM USAGE

Large Language Models (LLMs) were used exclusively to aid in the writing and polishing of this manuscript. Specifically, we employed an LLM to refine language, improve readability, and ensure clarity in various sections of the paper. The model assisted with tasks such as sentence rephrasing, grammar checking, and enhancing the overall flow of the text.

It is important to note that the LLM was not involved in the ideation, research methodology, or experimental design. All research concepts, ideas, and analyses were solely developed and conducted by the authors. The contributions of the LLM were limited to improving the linguistic quality of the paper, without involvement in scientific content or data analysis.

We further ensured that all LLM-generated text adheres to ethical guidelines and does not contribute to plagiarism or scientific misconduct.

## A.4   HESSIAN–VECTOR PRODUCT APPROXIMATION FOR A NEURON SET

During pruning, we consider removing a set of FFN neurons $S$ in layer $\ell$. Let $\boldsymbol{W}_\ell$ denote the input projection matrix of this layer, and let $\hat{\boldsymbol{W}}_{\ell,S}$ be the concatenation of the row vectors $\{\boldsymbol{w}_i\}_{i\in S}$ to be zeroed. The induced loss change admits the second-order expansion (layer-local) around $\boldsymbol{W}^*$:

$$\left|\Delta\mathcal{L}_{\ell,S}\right| \approx \left|\sum_{i\in S} \boldsymbol{w}_i^\top \nabla_{\boldsymbol{w}_i}\mathcal{L} \ - \ \frac{1}{2}\sum_{i,j\in S} \boldsymbol{w}_i^\top \boldsymbol{H}_{\ell\ell}\,\boldsymbol{w}_j\right|, \qquad \boldsymbol{H}_{\ell\ell} \equiv \nabla^2_{\boldsymbol{W}_\ell}\mathcal{L}(\boldsymbol{W}^*). \tag{A.1}$$

By the triangle inequality, we obtain an upper bound. We use this upper bound to denote the loss change induced by the removal of a neuron set, and take it as the importance score for guiding pruning decisions.

$$\left|\Delta\mathcal{L}_{\ell,S}\right| \ \leq \ \left|\sum_{i\in S} \boldsymbol{w}_i^\top \nabla_{\boldsymbol{w}_i}\mathcal{L}\right| \ + \ \frac{1}{2}\left|\sum_{i,j\in S} \boldsymbol{w}_i^\top \boldsymbol{H}_{\ell\ell}\,\boldsymbol{w}_j\right|. \tag{A.2}$$

Define the aggregated direction $\boldsymbol{w}_S := \sum_{j\in S} \boldsymbol{w}_j$. The quadratic term can be rewritten *exactly* as

$$\sum_{i,j\in S} \boldsymbol{w}_i^\top \boldsymbol{H}_{\ell\ell}\,\boldsymbol{w}_j \ = \ \boldsymbol{w}_S^\top \boldsymbol{H}_{\ell\ell}\,\boldsymbol{w}_S \ = \ \sum_{i\in S} \boldsymbol{w}_i^\top\left(\boldsymbol{H}_{\ell\ell}\,\boldsymbol{w}_S\right). \tag{A.3}$$

Hence, for a given set $S$, the per-neuron second-order contribution is $\boldsymbol{w}_i^\top (\boldsymbol{H}_{\ell\ell} \boldsymbol{w}_S)$, and Eq. A.1 can be evaluated via a single Hessian–vector product (HVP) with vector $\boldsymbol{w}_S$.

In practice, when ranking neurons *before* selecting $S$, the joint direction $\boldsymbol{w}_S$ is unknown. Following the structural-direction idea of Nonnenmacher et al. (2022), we approximate $\boldsymbol{w}_S$ by a layer-wise representative direction $\boldsymbol{w}_{\text{struc}}$, constructed by concatenating all row vectors of the layer's input projection matrix into a single long vector. This design captures the structural coupling within the layer and allows us to approximate the second-order interactions without explicitly computing the full Hessian.

$$\sum_{i,j \in S} \boldsymbol{w}_i^\top \boldsymbol{H}_{\ell\ell} \boldsymbol{w}_j \approx \left(\sum_{i \in S} \boldsymbol{w}_i^\top\right)\left(\boldsymbol{H}_{\ell\ell} \boldsymbol{w}_{\text{struc}}\right) \tag{A.4}$$

Now, applying the triangle inequality again yields a per-neuron decomposable upper bound:

$$\left|\Delta\mathcal{L}_{\ell,S}\right| \lesssim \sum_{i \in S} \left|\boldsymbol{w}_i^\top \nabla_{\boldsymbol{w}_i}\mathcal{L}\right| + \frac{1}{2}\sum_{i \in S}\left|\boldsymbol{w}_i^\top\left(\boldsymbol{H}_{\ell\ell}\,\boldsymbol{w}_{\text{struc}}\right)\right|. \tag{A.5}$$

We therefore define the Taylor-based importance score for a single neuron $i$ as:

$$\Delta\mathcal{L}_i^{\text{Taylor}} := \left|\boldsymbol{w}_i^\top \nabla_{\boldsymbol{w}_i}\mathcal{L}\right| + \frac{1}{2}\left|\boldsymbol{w}_i^\top\left(\boldsymbol{H}_{\ell\ell}\,\boldsymbol{w}_{\text{struc}}\right)\right|, \tag{A.6}$$

where both terms are averaged over the calibration dataset. This avoids materializing $\boldsymbol{H}_{\ell\ell}$ while capturing layer-local curvature through a single HVP, and ensures that $\sum_{i \in S} \mathcal{I}_i^{\text{Taylor}}$ upper-bounds the loss change for set $S$.

