# OpenReview forum: "Beyond Taylor Expansion: Intermediate Activation Perspectives in Structured Pruning"
_ICLR.cc/2026/Conference — ICLR 2026 Conference Withdrawn Submission_

### Official Review · Reviewer_FthH · 2025-10-31

**Soundness:** 2
**Presentation:** 3
**Contribution:** 2
**Rating:** 4
**Confidence:** 4

**Summary:**

This paper proposes a new standard for structured pruning in large language models, called ActTaylor. The authors first highlight a key limitation of existing pruning methods based on Taylor expansion: these methods only measure the parameters’ backward sensitivity to the loss, while ignoring the actual forward utilization of neurons during propagation. As a result, neurons with high Taylor scores may have activation values that remain near zero most of the time, contributing very little to the model’s real performance. To address this issue, ActTaylor introduces a hybrid importance score that combines two key factors through a geometric mean called the loss sensitivity from the Taylor expansion Ti and the statistical activation matrix Ai, which represents forward utilization. The proposed method enables one-shot pruning without any retraining. Experimental results show that on LLaMA-2 7B/13B and Mistral-7B models, ActTaylor consistently and significantly outperforms state-of-the-art structured pruning baselines such as LLM-Pruner and SliceGPT.

**Strengths:**

The motivation of the paper is very clear. The authors accurately identify the core limitation of existing Taylor expansion–based methods as the disconnection between loss sensitivity and forward utilization, and the analysis presented in Section 3.3 is highly convincing. Moreover, the authors propose a principled solution based on a clear assumption — that a neuron must be important in both backward sensitivity and forward utilization.

The proposed method consistently outperforms existing state-of-the-art structured pruning approaches across multiple models (e.g., LLaMA-2 7B/13B, Mistral-7B) and benchmarks. On LLaMA-2 13B, the model retains performance within only 3% of the original even under a 20% pruning ratio.

**Weaknesses:**

1. The core idea of the paper is to perform pruning by combining Taylor expansion with activation information. A natural baseline for comparison would therefore be weight–activation–based methods, such as Wanda. Although the paper compares its approach with state-of-the-art methods like LLM-Pruner which is gradient-based and SliceGPT which is decomposition-based, it lacks comparisons with conceptually closer and simpler baseline, particularly those that explicitly incorporate activation information, such as Wanda and RIA.
2. Appendix A.4 derives the Ti score, where a layer-wise Wstruc is used to approximate the HVP. This is a strong approximation. Although the results are promising, it would be helpful if the authors could provide more discussion or explanation regarding the trade-off between computational cost and accuracy of this approximation.
3. Regarding the balancing coefficient λ and the activation moment order p, the paper mentions that p=4 and λ is determined through grid search. However, no analysis is provided on the sensitivity of these hyperparameters. For example, why is the 4th-order moment chosen instead of the 1st-order which mean amplitude or 2nd-order indicate energy? How significantly does the choice of λ affect the results? It would be valuable if the authors could visualize these effects to provide a more intuitive understanding of the parameter sensitivity.

**Questions:**

1. Could the authors provide a sensitivity analysis of the hyperparameters λ and p? For example, how does the model perform when using the 1st-order moment p=1 or 2nd-order moment p=2 instead?
2. I am curious why the authors chose to combine the Taylor score and activation using a multiplicative geometric mean. How would an additive combination perform and could it potentially yield better results?
3. How does the proposed method perform under higher pruning ratios such as 50%? Could the authors provide additional comparisons with activation-guided pruning methods such as Wanda to further support the results?

---

### Official Review · Reviewer_6Us6 · 2025-10-31

**Soundness:** 2
**Presentation:** 2
**Contribution:** 2
**Rating:** 4
**Confidence:** 4

**Summary:**

This paper introduces ActTaylor, an activation-enhanced Taylor criterion for structured pruning of large language models (LLMs). Traditional pruning approaches estimate the loss increase from removing parameters using first- or second-order Taylor expansions, but these approximations ignore the hierarchical dependencies of deep networks—specifically, how a neuron’s activation propagates information forward. To address this, ActTaylor augments the Taylor importance score with moment statistics of intermediate activations. The final neuron importance (Eq. 8) is computed as a geometric mean between the Taylor-based loss sensitivity and the p-th moment of its activation. Experiments on LLaMA-2 (7B, 13B) and Mistral 7B under 20% and 30% structured pruning show consistent improvements over prior work, including LLM-Pruner (NeurIPS 2023), FLAP (AAAI 2024), SliceGPT (ICLR 2024), and SVD-LLM (ICLR 2025). ActTaylor achieves up to +14 pp improvement in average accuracy on reasoning benchmarks, all in a one-shot (no retraining) setting.

**Strengths:**

1. Clear motivation and mathematical consistency: The limitation analysis in §3.3 (Eq. 7 + Figure 1) convincingly shows why Taylor expansion misjudges inactive neurons.

2. Simple yet effective criterion: Eq. (8) requires minimal computation and fits neatly into existing pruning pipelines.

3. Strong empirical results: Table 1 shows ActTaylor surpassing all baselines across 8 tasks and 3 models, without retraining.

4. Scalable implementation: Using HVP approximations (Appendix A.4) and a 256-sample calibration dataset is computationally practical.

**Weaknesses:**

1. Limited novelty: ActTaylor primarily combines Taylor + activation statistics, which is a straightforward heuristic rather than a new theoretical advance beyond second-order Taylor criteria.

2. Missing recent baselines: Lacks comparison with Wanda (Sun et al., ICLR 2024), SOSP (ICLR 2022), and activation-sensitive methods like SOLA (AAAI 2025b). This makes claims of state-of-the-art performance incomplete.

3. Insufficient analysis of hyperparameters: No sensitivity plot for λ or p in Eq. (8); results might depend heavily on calibration dataset statistics.

4. No structured sparsity verification: The paper claims “hardware-friendly” pruning but never reports FLOPs reduction, throughput, or wall-clock speedups.

5. Visualization gaps: Figures 1 and 2 are informative but omit distribution of importance scores or cumulative retained energy plots that could clarify why ActTaylor helps.

6. Lack of downstream finetuning results: The work claims “one-shot pruning” is sufficient, yet showing post-tuning recovery could strengthen conclusions.

**Questions:**

1. λ and p sensitivity: How do different λ values affect the trade-off between activation and Taylor terms?
Would an adaptive λ per layer outperform a fixed global one?

2. Forward vs backward correlation: Did you measure correlation between activation magnitude and Taylor sensitivity across layers (e.g., Pearson r)?

3. Scalability: How does computational cost scale with model size and dataset size for computing HVPs?

4. Hardware impact: Can you provide latency or FLOPs reductions achieved after structured pruning?

5. Comparisons with SOLA (AAAI 2025) or DISP-LLM (NeurIPS 2024)? Both incorporate activation sparsity and could serve as stronger baselines.

---

### Official Review · Reviewer_QEZ6 · 2025-11-02

**Soundness:** 2
**Presentation:** 2
**Contribution:** 2
**Rating:** 4
**Confidence:** 4

**Summary:**

The paper introduces ActTaylor, a criterion for structured pruning of Transformer FFN neurons that augments classical (first/second‑order) Taylor loss‑sensitivity with intermediate activation statistics to better reflect a neuron’s forward contribution. The core idea is to score each neuron by the geometric mean of its Taylor-based importance and the p‑th moment of its activation over a small calibration set, so a neuron must be both loss‑sensitive and actively used to be retained. The method uses an efficient second‑order approximation via a single Hessian–vector product against a layer‑level “structural” direction, enabling one‑shot pruning without fine‑tuning. Motivation comes from the observation that many FFN neurons are near‑inactive across layers (see the activation distributions in the figure on p.5), which Taylor-only scoring can overvalue because it ignores forward utilization. Empirically, across LLaMA‑2 7B/13B and Mistral‑7B, ActTaylor outperforms structured‑pruning baselines at 20% and 30% sparsity on WikiText2 perplexity and eight reasoning benchmarks; e.g., average accuracy gains over the best baseline are +7.8% / +12.9% for LLaMA‑2‑7B and +12.5% / +14.0% for LLaMA‑2‑13B (see Table1 on p.7 and ablations on p.8).

Contributions

1. Activation‑enhanced importance score for structured pruning. Formalizes neuron importance as (I_i = A_i^{\lambda} \cdot T_i^{1-\lambda}), combining activation moments with Taylor loss‑sensitivity to jointly capture forward utilization and backward sensitivity.

2. Efficient second‑order criterion and practical one‑shot procedure. Derives a decomposable upper bound and computes the quadratic term with a single HVP using a layer‑wise structural direction, enabling scalable, training‑free pruning of FFN neurons.

3. Strong empirical validation. On LLaMA‑2 and Mistral, ActTaylor consistently beats recent structured‑pruning baselines (LLM‑Pruner, FLAP, SliceGPT, SVD‑LLM) at 20–30% sparsity on perplexity and commonsense benchmarks; the paper highlights widespread near‑zero activations across depths (figure p.5), a comprehensive results table (p.7), and ablations showing lower perplexity than Taylor‑only scoring as sparsity increases (p.8).

**Strengths:**

* Originality.
The paper reframes structured pruning by explicitly combining backward loss sensitivity with forward utilization via activation statistics. The importance score (I_i = A_i^{\lambda} \cdot T_i^{1-\lambda}) is a simple but fresh formulation that addresses a blind spot in Taylor-only criteria: Taylor coefficients depend on gradients and curvature but not on the actual activation (h_i), so they can overvalue neurons that rarely fire. The text makes this limitation precise and supports it with evidence, then proposes a geometric integration that requires a neuron to be both sensitive and used. The motivation is well argued in “Taylor expansion vs. forward influence,” and the activation distributions in Figure 1 (p. 5) visually ground the claim that many FFN neurons are near-inactive across depths, which justifies the activation-enhanced perspective.

* Quality.
Methodologically, the work is careful: it derives a per‑neuron second‑order criterion using a single Hessian–vector product against a layer‑level structural direction, giving a decomposable upper bound that is practical at LLM scale (Appendix A.4, Eqs. A.1–A.6). This is a reasonable curvature approximation that preserves the spirit of second‑order pruning without the usual computational blow‑up. The experimental design is also sound: calibration uses a modest C4 subset, evaluation covers WikiText2 perplexity plus eight reasoning benchmarks, and comparisons include relevant structured‑pruning and decomposition baselines. Ablations (Figure 2, p. 8) show the proposed score maintains lower perplexity than Taylor‑only as sparsity increases, which is exactly the failure mode the method claims to fix. The end‑to‑end procedure is one‑shot (no fine‑tuning or compensation), and the results table (Table 1, p. 7) shows consistent wins at both 20% and 30% sparsity across LLaMA‑2 7B/13B and Mistral‑7B.

* Clarity.
The paper reads cleanly. The limitation of Taylor criteria is explained with equations that connect gradients/curvature to missing dependence on activations, plus an illustrative example that makes the mismatch concrete. The proposed score is stated crisply (Eq. 8, p. 6), the layer‑wise sparsity schedule is explicit (Eq. 9), and the appendix walks through the HVP approximation step by step. Figures and tables are placed where they answer natural questions: Figure 1 motivates activation sparsity; Table 1 summarizes main outcomes; Figure 2 supports the ablation narrative. Implementation choices and evaluation protocol (datasets, harness, and shot settings) are specified, and an anonymous code link is provided for reproducibility.

* Significance.
Practically, the method improves the core trade‑off that matters for deployment: it yields better accuracy at the same structured sparsity and does so with a training‑free, hardware‑friendly pruning unit (whole FFN neurons). The reported gains are sizable and broad, for example average accuracy improvements over the best baseline of +7.8%/+12.9% at 20%/30% sparsity on LLaMA‑2‑7B and +12.5%/+14.0% on LLaMA‑2‑13B, while staying near the unpruned model at moderate sparsity (Table 1, p. 7). Conceptually, the work nudges the pruning community to balance backward signals with forward usage, a perspective that should transfer to other structured units beyond FFN neurons. Given how much inference cost still bottlenecks LLM adoption, a simple, scalable criterion that consistently improves accuracy at fixed compute cuts is likely to influence both research baselines and practical compression stacks.

**Weaknesses:**

1. Positioning vs prior art using activations is underdeveloped.
   The central novelty is the geometric combination of a Taylor score with activation moments. But the related work already notes methods that leverage activations or “soft activation sparsity” for compression (e.g., Wanda and SOLA) and several Taylor-style structured pruners. The paper does not include direct baselines or strong ablations that isolate what the activation term buys over the simplest activation-driven structured heuristics. Concretely, there is no comparison to: (i) an activation‑only structured score (rank by (A_i)); (ii) a forward‑magnitude proxy that actually reflects the FFN contribution ( |W_{2,:,i}|\cdot A_i ); or (iii) additive vs multiplicative combinations with matched hyperparameters. Adding these would clarify where the gain comes from and how much the geometric integration matters beyond reweighting. Table 1 compares against Taylor and decomposition baselines, but not activation-only structured baselines.
   Actionable: add baselines (i)–(iii) above at 20/30/40% sparsity on LLaMA‑2 7B/13B and Mistral‑7B, keeping the same calibration and evaluation settings as Table 1 to isolate the effect.

2. Forward influence is stated as (W_{2,:,i},h_i), yet (W_2) is not represented explicitly in the activation term.
   Section 3.3 argues the neuron's forward contribution scales with (|h_i|) through (W_{2,:,i} h_i), but the proposed importance uses (A_i) (activation moments) and a Taylor term w.r.t. the input row (w_i) only. There is no explicit factor that up‑ or down‑weights by the output column norm or sensitivity of (W_{2,:,i}). This can mis-rank neurons whose activations are moderate but whose output columns are large, or vice versa.
   Actionable: evaluate two variants: (I_i = ( |W_{2,:,i}|\cdot A_i)^{\lambda}, T_i^{1-\lambda}) and (I_i = A_i^{\lambda}, (T_i \cdot |W_{2,:,i}|)^{1-\lambda}). Report rank‑correlations and downstream performance vs the current (I_i).

3. The ReLU illustrative example is shaky and distracts from the real argument.
   The text claims Taylor coefficients “need not vanish” even when (h_i=0) and illustrates with ReLU at (a_i\le 0). For ReLU with (a_i<0), (\sigma'(a_i)=0) so first‑order terms vanish; at exactly (a_i=0) is a measure‑zero corner case. The broader point about smooth saturating gates (GELU/SwiGLU) is valid, but the ReLU example as written risks confusion.
   Actionable: replace with a GELU/SwiGLU example where (\sigma'(a_i)) is small but nonzero while (h_i) remains near zero, or explicitly caveat the ReLU case. A short numeric toy example would help.

4. “Upper bound” language becomes ambiguous after the structural‑direction approximation.
   Appendix A.4 derives an upper bound using the true joint direction (w_S), then replaces it with a layer‑wise structural direction (w_{\text{struc}}) to get a decomposable per‑neuron score. After this substitution, the formal upper‑bound guarantee generally no longer holds, yet the prose still implies a bound‑like interpretation. Clarifying this matters because the method’s attractiveness rests on the one‑shot safety of the score.
   Actionable: either (a) provide a lemma that quantifies the distortion introduced by (w_{\text{struc}}) (e.g., a constant‑factor or correlation guarantee under mild assumptions), or (b) drop “upper bound” phrasing for the approximate score and add an empirical validation: measure the true second‑order loss change for small random sets and report correlation vs the summed per‑neuron scores.

5. Limited analysis of hyperparameters and sensitivity.
   The paper fixes (p=4) and tunes (\lambda) on C4 but provides no sensitivity curves or evidence that conclusions are stable across models and calibration sets. Given (I_i = A_i^{\lambda}T_i^{1-\lambda}) is controlled by these choices, this is a major missing piece.
   Actionable: include heatmaps for (\lambda \in [0,1]) and (p\in{1,2,4,8}) showing perplexity and average task accuracy at 20/30% sparsity, for at least LLaMA‑2‑7B. Report the (\lambda) chosen per model and whether a single (\lambda) transfers.

6. Calibration dependence and domain shift are untested.
   All criteria are computed on 256 C4 samples of length 4096, while evaluation spans WikiText2 and eight downstream tasks. The paper does not test whether the ranking is robust to calibration domain, sequence length, or number of samples.
   Actionable: vary calibration source (C4 vs WikiText2), sequence length (2k/4k/8k), and sample count (64/128/256/512). Plot performance vs compute to justify the chosen setting and show robustness.

7. Baselines: coverage and fairness.
   LLM‑Pruner is omitted for Mistral (“not applicable”), and there is no comparison to other second‑order structured methods like LLM Surgeon, which the related work highlights. Moreover, methods that use compensation or iterative pruning should be run in their recommended regimes with matched or reported compute. The paper reports that its own HVPs use four H100s, but gives no wall‑clock or FLOPs for either ActTaylor or baselines.
   Actionable: (i) add LLM Surgeon where applicable, or explain concretely why it cannot be run; (ii) document baseline configurations, recovery steps, and compute budgets; (iii) report pruning‑time FLOPs, wall‑clock, and peak memory for all methods to substantiate “efficient and scalable.”

8. Hardware benefits are claimed but not measured.
   The text motivates structured pruning by hardware efficiency, yet there are no latency or throughput results, only perplexity/accuracy. Without speedups and memory savings, the practical significance is ambiguous.
   Actionable: report tokens/sec and VRAM usage for the pruned models at batch sizes {1, 8, 32} and seq lengths {1k, 4k, 8k} on a standard GPU, plus end‑to‑end latency for generation. Include roofline-style plots showing realized vs theoretical FLOP reductions.

9. Scope limited to FFN neurons; consequences not explored.
   Only FFN units are pruned. The paper does not analyze residual attention bottlenecks or whether head/channel pruning could be integrated with the same activation‑enhanced principle. This matters for achieving speedups beyond moderate levels.
   Actionable: add a small study that combines ActTaylor‑FFN with a lightweight attention‑head/channel pruning rule, or at least quantify the fraction of layer FLOPs removed vs the realized speedups to set expectations.

10. Ablations focus on perplexity, not downstream tasks, at higher sparsity.
    Figure 2 extends perplexity ablations to 40% but downstream results are shown only at 20% and 30%. It’s unclear whether the claimed stability persists on tasks as sparsity grows.
    Actionable: report downstream accuracy at 35–40% sparsity for at least one model, and include failure‑mode breakdowns (which tasks collapse first, and why).

**Questions:**

1. Positioning vs activation‑based compression.
   Ask. Where exactly does ActTaylor’s novelty sit relative to activation‑driven heuristics and “soft activation sparsity” approaches cited in Related Work, and how much of the gain is due to using activations at all versus the geometric combination in Eq. 8?
   Why it matters. Figure 1 on p. 5 argues many FFN neurons are near‑inactive; without direct activation‑only structured baselines, it’s hard to separate the effect of activation statistics from the specific Taylor integration.
   What would help. Add baselines that prune by: (i) activation‑only (A_i); (ii) a forward‑magnitude proxy (|W_{2,:,i}| \cdot A_i); and (iii) additive vs multiplicative blends with matched hyperparameters. Report at 20/30/40% on LLaMA‑2 7B/13B and Mistral‑7B with the same calibration and schedule used in Table 1 (p. 7).

2. Role of the output matrix (W_2).
   Ask. Section 3.3 states a neuron’s forward contribution scales through ((W_2)*{:,i} h_i), yet (W_2) does not explicitly enter the importance score in Eq. 8. Why is this omission benign?
   Why it matters. Two neurons with similar activations but very different (|(W_2)*{:,i}|) could be mis‑ranked.
   What would help. Provide ablations with (I_i = (|W_{2,:,i}|,A_i)^\lambda T_i^{1-\lambda}) and (I_i = A_i^{\lambda}(T_i|W_{2,:,i}|)^{1-\lambda}), plus rank‑correlation vs the current score and downstream performance deltas.

3. Status of the “upper bound” after the structural‑direction substitution.
   Ask. In Appendix A.4, the per‑set upper bound becomes per‑neuron and decomposable after replacing (w_S) with the layer‑wise (w_{\text{struc}}). Does the bound still hold, or is it now an approximation? Please state this precisely.
   Why it matters. The efficiency and one‑shot safety pitch hinges on the bound interpretation.
   What would help. Either a lemma quantifying the distortion introduced by (w_{\text{struc}}) or an empirical study: sample small neuron sets S, compute the true second‑order loss change vs (\sum_{i\in S} I_i^{\text{Taylor}}), and report correlation/Calibration‑AUC.

4. ReLU illustrative example.
   Ask. The “Illustrative example” on p. 5 implies Taylor coefficients need not vanish even when (h_i=0) for ReLU. At (a_i<0), (\sigma'(a_i)=0), so first‑order terms vanish, and the (a_i=0) corner case is measure‑zero. Can you revise this example?
   Why it matters. The point is valid for smooth saturating gates (GELU/SwiGLU), but the current ReLU wording is potentially misleading.
   What would help. Swap in a GELU/SwiGLU numeric toy demonstrating near‑zero (h_i) with nonzero sensitivity, or clearly caveat the ReLU case.

5. Hyperparameter sensitivity of (\lambda) and (p).
   Ask. Why is (p=4) fixed and how sensitive are results to (\lambda) and (p) across models?
   Why it matters. Eq. 8 depends critically on (\lambda) and the chosen moment. Currently (\lambda) is tuned on C4; robustness is unclear.
   What would help. Provide heatmaps over (\lambda\in[0,1]) and (p\in{1,2,4,8}) for LLaMA‑2‑7B at 20/30% sparsity, showing perplexity and averaged task accuracy. Indicate whether a single (\lambda) transfers to all models.

6. Calibration data dependence and sequence length effects.
   Ask. How stable are rankings under calibration dataset/domain, sequence length, and sample count?
   Why it matters. You compute criteria on 256 samples from C4 at 4096 tokens (p. 6). Evaluation spans WikiText2 and multiple tasks; domain and length shifts may matter.
   What would help. Vary calibration source (C4 vs WikiText2), length (2k/4k/8k), and sample count (64/128/256/512). Plot performance vs compute, justifying the chosen 256×4096 setting. Also clarify padding/mask handling when aggregating (A_i).

7. Efficiency accounting and baseline parity.
   Ask. What are the pruning‑time FLOPs, wall‑clock, and peak memory for ActTaylor vs each baseline, and are baselines run in their recommended regimes with matched compute?
   Why it matters. You mention HVPs on four H100s, but no cost table; Table 1 compares quality only. Reproducibility depends on fair compute budgets.
   What would help. Add a small table with pruning‑time FLOPs, hours, and max memory for all methods and models; document exact baseline configs and any compensation or iterative steps used.

8. Hardware speedups.
   Ask. What realized speedups do the pruned models achieve on standard hardware?
   Why it matters. The motivation emphasizes hardware friendliness of structured pruning, but no latency/throughput results are reported.
   What would help. Report tokens/sec and VRAM for batch sizes {1, 8, 32} and context lengths {1k, 4k, 8k}, plus end‑to‑end generation latency, and compare realized vs theoretical FFN FLOP reduction.

---

### Note · Authors · 2026-01-02

I have read and agree with the venue's withdrawal policy on behalf of myself and my co-authors.